# VAV Proteins as Double Agents in Cancer: Oncogenes with Tumor Suppressor Roles

**DOI:** 10.3390/biology10090888

**Published:** 2021-09-08

**Authors:** Myriam Cuadrado, Javier Robles-Valero

**Affiliations:** 1Centro de Investigación del Cáncer, CSIC-University of Salamanca, 37007 Salamanca, Spain; mcuadrado@usal.es; 2Centro de Investigación Biomédica en Red de Cáncer (CIBERONC), CSIC-University of Salamanca, 37007 Salamanca, Spain

**Keywords:** RHO GEFs, VAV proteins, oncogene, tumor suppressor, RAC1, NFAT, NOTCH1, T-ALL, PTCL, lung cancer, signaling, mouse models

## Abstract

**Simple Summary:**

The role of the VAV family (comprised of VAV1, VAV2, and VAV3) in proactive pathways involved in cell transformation has been historically assumed. Indeed, the discovery of potential gain-of-function VAV1 mutations in specific tumor subtypes reinforced this functional archetype. Contrary to this paradigm, we demonstrated that VAV1 could unexpectedly act as a tumor suppressor in some in vivo contexts. In this review, we discuss recent findings in the field, where the emerging landscape is one in which GTPases and their regulators, such as VAV proteins, can exhibit tumor suppressor functions.

**Abstract:**

Guanosine nucleotide exchange factors (GEFs) are responsible for catalyzing the transition of small GTPases from the inactive (GDP-bound) to the active (GTP-bound) states. RHO GEFs, including VAV proteins, play essential signaling roles in a wide variety of fundamental cellular processes and in human diseases. Although the most widespread archetype in the field is that RHO GEFs exert proactive functions in cancer, recent studies in mice and humans are providing new insights into the in vivo function of these proteins in cancer. These results suggest a more complex scenario where the role of GEFs is not so clearly defined. For example, VAV1 can unexpectedly play non-catalytic tumor suppressor functions in T-cell acute lymphoblastic leukemia (T-ALL) by controlling the levels of the active form of NOTCH1 (ICN1). This review focuses on emerging work unveiling tumor suppressor roles for these proteins that should prompt a reevaluation of the role of VAV GEF family in tumor biology.

## 1. Introduction

RHO GTPase pathways are involved in a wide range of cellular processes, such as proliferation, cell survival, and apoptosis, as well as cell-specific roles in many tissues regulating immune response, angiogenesis, and neurogenesis [1]. Thus, alterations of these pathways contribute to a large variety of relevant pathologies, including malignant transformation and cancer progression [1,2]. Historically, it has been assumed that the main mechanism of deregulation of RHO GTPases and their regulators in cancer was through changes in the expression levels [1,3,4]. Although this paradigm remains valid in some tumors, this idea has been challenged with the development of deep sequencing technologies, as some members of the RHO family (RAC1, RHOA, and CDC42) and their regulators (RHO GDP/GTP exchange factors (GEFs), such as P-REX2 and VAV1) [5] have been found mutated in a large number of human tumors [1]. Furthermore, the detection of not only gain-of-function but also loss-of-function mutations in RHO GTPase-related genes underscores the complexity of the RHO GTPase routes in human tumors.

One of the best examples of this complex picture in the RHO GTPase field is the VAV GEF family, a group of versatile signal transducers that work as GEFs for RHO GTPases as well as adaptor molecules [6,7]. This family has three members (VAV1, VAV2, and VAV3) that show overlapping but not identical expression patterns. While VAV1 is mainly expressed in hematopoietic cells, VAV2 and VAV3 are also found in non-hematopoietic tissues [6,7]. The most characteristic catalytic function of VAV proteins involves the stimulation of GDP/GTP exchange, a step that favors the transition of the RHO GTPases from the inactive (GDP-bound) to the active, GTP bound state [6]. VAV proteins use RAC1 as a main substrate and, to a much lower extent, RHOA [8,9]. The activation of RHO GTPases leads to the remodeling of the F-actin cytoskeleton and the stimulation of downstream elements, such as the c-Jun N terminal kinase (JNK) and the transcriptional factors AP1 and serum responsive factor (SRF) [6]. They all are characterized by a highly complex structure composed of different domains that perform functions related to: (a) the intramolecular regulation of the protein activity (CH, Ac, SH3, and SH2 domains; Figure 1A). (b) The catalytic activation of RAC1 and other related GTPases (DH, PH, and ZF regions; Figure 1A). (c) Adaptor-like functions, such as the interaction with a variety of proline rich region (PRR) proteins (SH3 domains; Figure 1A) and the activation of a phospholipase Cγ1 (PLCγ1)-dependent pathway that leads to the stimulation of the nuclear factor of activated T cells (NFAT) (CH region; Figure 1A) [6,10]. Most of these adaptor functions are not related to the catalytic activity of VAV proteins [6,7].The biological activity of VAV proteins is controlled by phosphorylation-dependent conformational changes (Figure 1B). Thus, in non-stimulated cells, nonphosphorylated VAV proteins are inactive in an autoinhibited conformation mediated by intramolecular interactions between both the CH-Ac region and the CSH3 domains with the central cassette DH-PH-ZF (Figure 1A,B). Upon phosphorylation of specific tyrosine residues located on the Ac, ZF, and CSH3 regions by either transmembrane or cytosolic protein tyrosine kinases (PTKs), this autoinhibited structure shifts towards an “open” and active conformation (Figure 1A,B). As a result, VAV proteins can exhibit a variety of catalysis-dependent and -independent biological responses being critical for the homeostasis of the nervous, cardiovascular, and immune systems [6]. These functions can be referred to as ‘canonical activities’, since they represent common and well-established features in the VAV family. In addition, the aberrant function of these proteins can contribute to several pathologies such as those related to the immune system and cancer. In particular, the VAV GEF family has been traditionally linked to protumorigenic actions in cancer [6]. This idea was reinforced by the use of both cancer cell lines and mouse models demonstrating the proactive role of VAV proteins in the development of different types of tumors, such as skin and breast cancer [11,12]. However, given the presence of structural domains that facilitate the interaction with a large number of protein partners and the particular features of some of the VAV-dependent pathways, it is conceivable that VAV proteins might antagonize cell transformation in certain in vivo contexts. In line with this, the discovery of the catalysis-independent tumor suppressor function of VAV1 buffering NOTCH1 signaling in T cells further hindered the understanding of the role of VAV proteins in cancer (Figure 1A,B) [5,13].

In the present review, we will discuss the recent findings reversing the long-held paradigm that VAV GEF proteins always favor tumor growth. In particular, we focus on the dual role of VAV1 in T cell neoplasms, with an emphasis on tumor cell-driven mechanisms that affect VAV1 control of NOTCH1 signaling.

## 2. VAV1 as a Tumor Suppressor in T-ALLs

Three decades ago, VAV1 was identified as an oncogene by the classical focus formation assays in NIH3T3 cells [14]. This connection with protumorigenic events continued to be supported by accumulative evidence from both cell lines and mouse models [6,11,12,15]. Despite this evidence, and perhaps counter-intuitively, the role of VAV1 in hematological malignancies did not appear to be significant. This view started to change upon the recent discovery of putative gain-of-function mutations in the VAV1 gene in human tumors. Indeed, recent genomic research using whole exome sequencing revealed that 2–17% of peripheral T-cell lymphomas (PTCLs) and 2–3% of lung tumor cases bear VAV1 mutations [16,17,18,19,20,21,22,23]. Given this background, it was quite surprising that VAV1-deficient mice develop T cell tumors at high frequency upon aging or when treated with different carcinogens [13,24]. This phenotype is driven exclusively by VAV1 deficiency, as compound VAV1−/−; VAV2−/−; VAV3−/− animals behave similarly to their VAV1−/− counterparts. The analysis of these tumor-bearing animals revealed the presence of abnormally high numbers of immature T cells in the thymus. Consistent with this, they develop T cell acute lymphoblastic leukemia (T-ALL), a type of cancer that arises from the malignant transformation of immature T lymphocytes lacking T cell receptor (TCR) expression [25]. Furthermore, flow cytometry analysis revealed that VAV1 could act as a tumor suppressor protein at different levels during T cell maturation. This result suggests a hitherto unknown function for VAV1, since it has been traditionally associated with TCR selection and antigen-dependent TCR signaling events in mature T cells [26]. Surprisingly, in silico analysis revealed a high degree of similarity of the VAV1−/− tumor transcriptome with that found deregulated in leukemic cells generated after ectopic expression of the intracellular domain of Notch1 (ICN1) in mouse bone marrow progenitors. In agreement with these data, transcripts commonly upregulated in Notch1-driven T-ALL such as *Hes1* and *Myc* were found in VAV1−/− tumor cells using qRT-PCR experiments. Overall, these data suggest a novel relationship between VAV1 and the NOTCH1 pathway. 

The strong oncogenic activity of NOTCH1, which is commonly mutated in more than 65% of human T-ALL cases, has emerged as a major regulator of T-ALL development [25]. NOTCH1 signaling is initiated upon binding of transmembrane ligands expressed on the surface of neighboring cells. Following this interaction, two successive proteolytic cleavages carried out by ADAM10 metalloprotease and γ-secretase complex, respectively, lead to migration of the intracellular domain (ICN1) to the nucleus and stimulation of its target genes involved in cell fate decision, metabolism and proliferation [25,27]. This transcriptional program ends with phosphorylation of the C-terminal PEST region, which targets ICN1 for FBXW7-mediated ubiquitination and degradation by the proteasome [28]. Most T-ALL-associated mutations result in the truncation of the PEST domain allowing ICN1 to evade proteasomal degradation. Therefore, NOTCH1 signaling is constitutively active in these types of tumors [29]. Our study revealed a defective ubiquitination of ICN1 in the absence of VAV1 in T cells, indicating a connection of VAV1 to the silencing step of NOTCH1 signaling through degradation [13]. Based on these results, we suspected that the best candidate for this regulatory step could be the E3 ubiquitin ligase CBL-B since it can bind to the SH3 C-terminal (CSH3) domain of VAV1 (Figure 1A) [30]. This novel adaptor function mediated by VAV1 through its C-terminal region favors the formation of cytosolic complexes between ICN1 and CBL-B and facilitates the CBL-B–mediated degradation of ICN1 in T cells by the proteasome. As a result, the ablation of VAV1 in mice leads to unbalanced Notch1 signaling, the activation of Notch1 target gene signatures, and the rapid emergence of T-ALL [5,13]. Further experiments demonstrated that the central catalytic core or the SH2 domain of VAV1 are not relevant in the formation of these complexes. Indeed, this new catalytic-independent function is still active in unstimulated T cells, demonstrating that VAV1–ICN1 connection is tyrosine phosphorylation-independent (Figure 1B) [5,13]. In line with this, we have observed that the engagement of this signaling mechanism does not require, unlike all the other catalysis-dependent and -independent functions so far known for VAV1 [6], a proper activation of the TCR or the prior phosphorylation of the protein, demonstrating that VAV1–ICN1 connection is tyrosine phosphorylation-independent [13]. According to these data, the VAV1–CBL-B suppressor pathway might be active in immature thymocytes (lacking TCR expression) as a way to ensure normal levels of ICN1 signaling during T cell maturation. All together, these results unveiled a new process for the regulation of ICN1 abundance in T lymphocytes that does not relies on its canonical degradation by FBXW7.

Finally, and most importantly, we also provide evidence supporting the idea that the downmodulation of this tumor suppressor route is important for the pathogenesis of human T-ALL patients. Among the various T-ALL oncogenic alterations reported to date, gain-of-function alterations in transcriptional factors such as LYL1, HOXA, TAL1, TLX1, and TLX3 represent a recurrent oncogenic hallmark of T-ALL [27]. In silico analyses of patient samples representative of these molecular T-ALL disease subtypes indicated that TLX+ T-ALL clinical subtype cases showed low abundance of VAV1 transcript as well as a similar gene signature to the murine VAV1−/− T-ALL cells [13]. Independent studies have demonstrated that TLX proteins are the most relevant oncogenic drivers for this subtype of T-ALL in humans [25,31]. Moreover, these proteins can directly repress the VAV1 gene, resulting in a downmodulation of the tumor suppressor role of the protein (Figure 2A). In line with this, primary cells directly obtained from TLX+ T-ALL patients showed much lower levels of VAV1 protein than patient-derived TLX– T-ALL cells. Consistent with these observations, the re-expression of VAV1 resulted in antiproliferative and pro-apoptotic effects in all the TLX+ cells tested. However, these effects could not be elicited when a VAV1 mutant protein incapable of binding CBL-B was used in the same experiments [13]. In line with the non-catalytical function of VAV1 described above, we also showed that the negative effect of ectopically expressed VAV1 in the fitness of TLX+ T-ALL cells could be recapitulated by the expression of a catalytically dead mutant of the protein unable to bind RHO GTPases. Taken together, these results highlighted the clinical significance of the VAV1–CBL-B suppressor pathway silencing in this leukemia subtype.

## 3. How Widespread Is This Tumor Suppressor Function in Neoplastic Processes?

As summarized above, this is the first time that the transcriptional repression of VAV1 gene appears to represent a key factor contributing to human cancer pathogenesis [13]. Contrary to this scenario, different cancer studies have reported unexpected expression of VAV1 in tissues where it is not normally expressed, such as pancreas and lung cancer [33,34]. This evidence suggests that the downregulation of the VAV1–CBL-B axis may not be relevant in other cancer types beyond T-ALL. However, the discovery of point mutations as well as gene fusions and truncations targeting the CSH3 domain of VAV1 predicts the importance of the VAV1 tumor suppressor-like pathway [35]. These alterations were found in both T cell neoplasms (mainly PTCLs) and lung tumors. According to these studies, the CSH3 region is the most commonly mutated hotspot of the protein (≈50% of the VAV1 mutations). Therefore, the alterations in the CSH3 domain, which is also responsible for the intramolecular inhibition of the protein (Figure 1A,B) [35], suggest an increase in the catalytic-dependent and independent outputs of the protein, as well as the prevention of the CBL-B–ICN1 complex formation favoring the development of human tumors. However, the relevance of these mutations from a functional and pathobiological perspective remains undefined. In this context, recent studies have shown that some VAV1 mutations targeting the CSH3 domain lead indeed to gain-of-function events [16]. However, these analyses have been limited to a small and overlapping subset of mutations targeting obvious regulatory layers of the protein. As a result, we do not know yet whether most VAV1 mutations found in human tumors act as bona fide oncogenic drivers in vivo and, if so, whether they do it autonomously or in combination with other genetic lesions.

In line with this, we decided to analyze the lymphomagenic potential of VAV1 mutations using an adoptive T cell transfer approach (J.R.-V., submitted paper). Our in vivo experiments indicated that a CSH3-terminally truncated protein that leads to the concurrent hyperstimulation of the RAC1 and NFAT pathways as well as the NOTCH1-derived signals is able to autonomously drive PTCL-like lymphomas in mice, specifically angioimmunoblastic T-cell lymphoma (AITL) subtype (J.R.-V., submitted paper) (Figure 2B). These animals exhibited a highly expanded population of CD4^+^ T cells with a T follicular helper (T_FH_) cell-like phenotype, a common feature found in previously reported AITL-associated cases in both mice and humans [36]. The contribution of VAV1 mutations in the development of T cell neoplasms in mice comes as no surprise. In a recent paper, it has been reported that transgenic mice expressing other oncogenic versions of VAV1 can develop GATA3^+^ PTCL-not otherwise specified–like (PTCL-NOS) tumors in the presence of *Trp53* deletion [37]. The lack of detection of this PTLC subtype in our adoptive transfer experiments is still unclear, although it might reflect the need of cooperating genetic events that could favor the emergence of a T cell subtype different from the T_FH_ phenotype characteristic of AITL (e.g., viral integrations in the case of adult T cell leukemia/lymphoma [ATLL], loss of *TP53* and/or *TET2* genes in AITL) [36,37,38]. The use of animal model-based experiments will help to understand the etiology of PTCL subtypes in the near future.

VAV CSH3 domain also facilitates the interaction with a large number of proteins such as the heterogenous nuclear ribonucleoprotein K (HNRNPK) and dynamin 2 (DNM2) (Figure 1A) [39,40,41]. We cannot exclude the possibility that some mutations preferentially prevent the binding of either HNRNPK or DNM2. The functional implication of this putative change in the spectrum of binding partners is currently unknown. However, it is worth noting that, similarly to CBL-B, HNRNPK and some DNM family members (e.g., DNM3) have been associated to tumor suppression activities [41,42]. Taken together, these proteins might play further adaptor-like functions in both normal and cancer cells, expanding the catalogue of VAV-dependent suppressor activities.

## 4. VAV2 and VAV3 in Cancer: Could They Also Be Involved in Tumor Suppressor Pathways?

As mentioned in the introduction section VAV2 and VAV3 show more widespread expression patterns than VAV1, which is primarily expressed in most hematopoietic lineages. Despite this relevant difference, VAV proteins share a very similar structure and can exhibit overlapping functions, suggesting that VAV2 and VAV3 might play tumor suppressor roles similar to those carried out by VAV1 beyond the long-held signaling archetype for RHO GEFs. At present, while the available in vivo data point to a clear oncogenic function of VAV2, this information clashes with our current knowledge about VAV1 described in this review, as well as the least studied gene of the family, VAV3. In line with this, some studies indicate that VAV3, but not VAV2, may trigger common VAV1-related pathways in cells. For example, it has been shown that VAV3 can stimulate the NFAT signaling pathway in T lymphocytes [43]. Although most of the catalytic functions take place in the cytosol, there are also data supporting the role of VAV1 and VAV3 in adaptor-like functions in the nucleus. These functions require the CSH3 domain of VAV1 and the DH-PH-ZF cassette of VAV3 [43,44,45]. The emergence of alternative VAV3-dependent functions suggests that it might also act as a tumor suppressor. Indeed, we have found that VAV3−/− mice develop certain tumors at higher frequencies than the control counterpart (M.C., unpublished data). Unlike the case of VAV1, there are no consistent examples of mutations for the VAV3 gene in cancer. This observation could suggest that the downregulation of this unexpected role for VAV3 in some specific tumor contexts could be driven by transcriptional repression of the gene rather than loss-of-function mutations. Whether this new function is similar to the VAV1-dependent suppressor pathway remains an open question to be addressed.

To date, there is no experimental evidence supporting these novel antitumorigenic functions for VAV2. Indeed, the generation of genetically engineered mice exhibiting varying amounts of VAV2 catalytic activity has demonstrated that the inhibition of the GEF activity prevents the generation of skin tumors in vivo [46]. In line with this, VAV2 was recently shown as frequently overexpressed in both cutaneous squamous cell carcinoma (cSCC) and head and neck squamous cell carcinoma (hnSCC) tumors [47]. The tumor effect induced by its overexpression is mediated by c-MYC- and YAP/TAZ-dependent transcriptional programs associated with regenerative proliferation and cell un-differentiation in these tumor types, respectively [48]. Taking together, the upregulation of VAV2 catalytic activity facilitates tumor development, similar to the effect caused by some VAV1 mutations in PTCLs [18].

Finally, it was reported that VAV2 along with VAV3 promote tumorigenesis and lung metastasis in breast cancer cells [11]. As a result, only the expression of both proteins could reverse these phenotypes caused by their double knockdown. However, there are specific functions in the maintenance of breast cancer epithelial structures that can be redundantly performed by either VAV2 or VAV3 [11]. Adding further evidence to the connection between Vav2 and VAV3 in cancer, genetic analysis has shown that VAV2 and VAV3 depletion decreases carcinogen-induced skin tumor formation while maintaining skin homeostasis in mice [12]. In any case, we cannot exclude the possibility at this moment that other VAV2- and/or VAV3-mediated pathobiological processes could utilize catalysis-independent pathways as seen in the case of VAV1 and other RHO GEFs [1]. Addressing this issue will require further studies using ad hoc-designed animal models.

## 5. Final Remarks

Despite the progress made in the understanding of the role of VAV proteins in cancer, the recent discovery of the tumor suppressor role of VAV1 has further highlighted the need to study the function of these proteins beyond their catalytic activity. This new scenario is supported by emerging evidence showing that many RHO GTPases and RHO GEFs could antagonize cell transformation under certain circumstances, while RHO GAPs may promote oncogenesis. For example, the most frequent RHOA alteration found in human T-cell lymphomas is a loss-of-function mutation targeting the Gly17 residue (RHOA^G17V^) in the GTP-binding domain [35]. The use of different animal models has clearly established the role of this dominant negative mutant in the lymphomagenesis process, suggesting that RHOA^G17V^ favors the activation of the ICOS–PI3K–AKT–mTOR axis [36,49,50]. Furthermore, the discovery of recurrent loss-of-function mutations in RHO GEF genes such as *ARHGEF10, ARHGEF10L,* and *PREX* [51,52] implicates these proteins in antitumorigenic activities. However, information about the functional effects of these mutations in the tumor biology is still lacking. Finally, the RHO GEF TIAM1 is another example of the complex function of RHO GTPase regulators in cancer, specifically in colorectal cancer [53], where it limits the acquisition of migratory properties through the inhibition of the YAP/TAZ pathway [53].

An important issue to be further addressed in the near future is to verify the relevance of the VAV1-dependent tumor suppressor function in VAV2- and VAV3-driven tumors, particularly in those that may be NOTCH1-dependent. At present, the biological link between VAV proteins and the NOTCH1 pathway has only been studied in the context of VAV1-dependent tumors. Indeed, the connection between VAV1 and CBL-B signaling pathways has previously been dissected in the context of the regulation of TCR-mediated downstream events during T cell activation [54]. However, it is possible that, as in the case of VAV1, the rest of the members of the VAV family proteins could be related to NOTCH1 signaling.

Another outstanding question is how this antitumorigenic effect attributable to VAV family proteins is functionally segregated or not from the canonical VAV-dependent pathways. In the case of VAV1, it is possible that the suppressor and canonical functions segregate depending on the differentiation state and TCR expression status of T lymphocytes. Given the reliance of VAV1 on tyrosine phosphorylation for the stimulation of canonical pathways, this functional transition can be established by the state of VAV1 phosphorylation reached at each stage of T cell maturation. In TCR– T cells, the suppressor pathway ensures the buffering of ICN1 signals, whereas in TCR+ T cells, optimal levels of NFAT and RAC1 stimulation are also reached upon VAV1-dependent phosphorylation. The functional boundaries between these two antagonistic functions can correlate with the physiological windows of NOTCH1 and canonical VAV1 signaling in T cell development [55]. Consistent with this idea, immature single-positive CD8^+^ cells (ISPCD8) are the final NOTCH1-dependent stage in T cell development and the latest one capable of undergoing transformation in VAV1−/− mice [13]. Further work will be needed to determine whether these two pathways can operate at the same time.

We also need to advance in the understanding of the role of VAV1 mutations both autonomously and in synergy with other genetic lesions found in PTCL and other tumors. In addition to VAV1 mutations, numerous studies have demonstrated the predominance of activating alterations in genes encoding proteins that participate in TCR signaling, being *PLCG1* and *CD28* the most frequently mutated genes found in this subtype of lymphomas [20,22,56]. Discerning the functional significance of these alterations in the context of VAV1 mutations, and in particular those that disrupt the tumor suppressor activity, could provide further insight into the exacerbated signaling that occurs in PTCLs. Indeed, this is an open question due to the limited number of mouse models that can adequately recapitulate the genetic lesions and pathobiological features found in PTCL patients. The adoptive transfer experiments (J.R.-V., submitted paper) and genetically modified animal models [37] could represent ideal tools for studying the functional significance of most of these alterations and ultimately testing therapeutical avenues for PTCLs.

Finally, the discovery of tumor suppressor activities of RHO GEFs, in particular, those related to VAV proteins, raises a new scenario for the development of approaches to inhibit the RHO GEF activities. Indeed, the functional duality of these proteins makes the therapeutic intervention a challenge to be solved in the coming years. It is worth noting that targeting the catalytic domains of RHO GEFs is to date believed to be the most feasible strategy for drug development in this field [57,58]. In this case, it is vital to understand all possible systemic consequences of RHO GEF inhibition. However, this is far from being an open-and-shut case. Interestingly, the GTPase-independent pathways could be conserved in cancer cells even under conditions of inhibition of catalytic activity. The use of mouse models and genetic approaches to unravel these unexpected signaling networks might help render VAV proteins druggable in the near future.

## 6. Conclusions

As summarized in this review, significant progress has been made in recent years in the understanding of the modus operandi of VAV proteins in cancer. Despite this, there are still pending questions, such as the organization of the VAV tumor promoter and suppression functions in different cancers, the functional and clinical significance of many of the alterations found in human tumors, and the pathways that contribute to cancer development. Future work should give us a holistic view of the contribution of GTPase-dependent and -independent pathways to the overall pathobiological program of each VAV protein. To this end, we will need to develop better animal models and experimental tools, such as patient-derived xenografts, organoids, and primary cancer cells. Finally, unraveling these questions will allow us to develop compounds that could be therapeutically effective.

## Figures and Tables

**Figure 1 biology-10-00888-f001:**
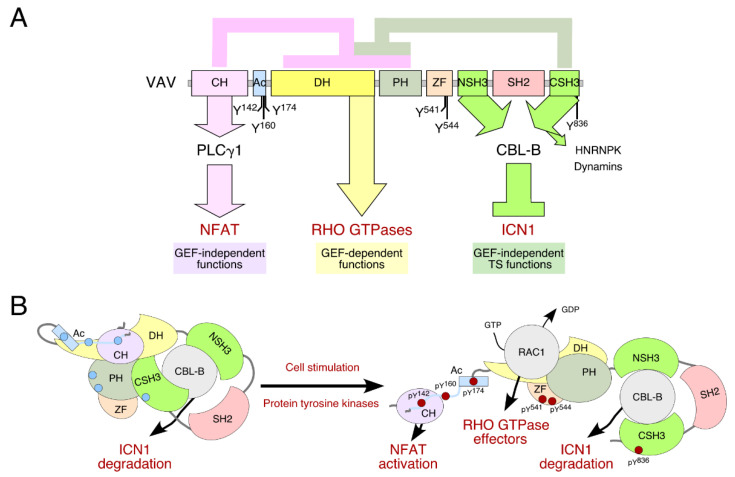
Structure and effector activities of VAV proteins. (**A**) Depiction of the structure of VAV proteins, the intramolecular interactions that control their biological activities (top), the main phosphosites involved in the activation step, and the main downstream pathways (bottom) activated by the indicated domains. The numbering refers to the sequence of VAV1. CH, calponin-homology; Ac, acidic; DH, Dbl-homology; PH, pleckstrin-homology; ZF, zinc finger; NSH3, most N-terminal SH3 domain; CSH3, most C-terminal SH3; TS, tumor suppressor. (**B**) Phosphorylation-mediated activation model of VAV proteins. The autoinhibited state of nonphosphorylated VAV proteins is stabilized by intramolecular contacts of the CH and CSH3 domains with the catalytic core DH-PH-ZF. Upon phosphorylation of tyrosine residues, the autoinhibited structure is released. Note that the suppressor activity of VAV that contributes to the control of NOTCH1 levels can be fully engaged by both the inactive and active versions of the protein. Nonphosphorylated and phosphorylated residues are shown as blue and red circles, respectively. Color codes for VAV domains are those used in panel A.

**Figure 2 biology-10-00888-f002:**
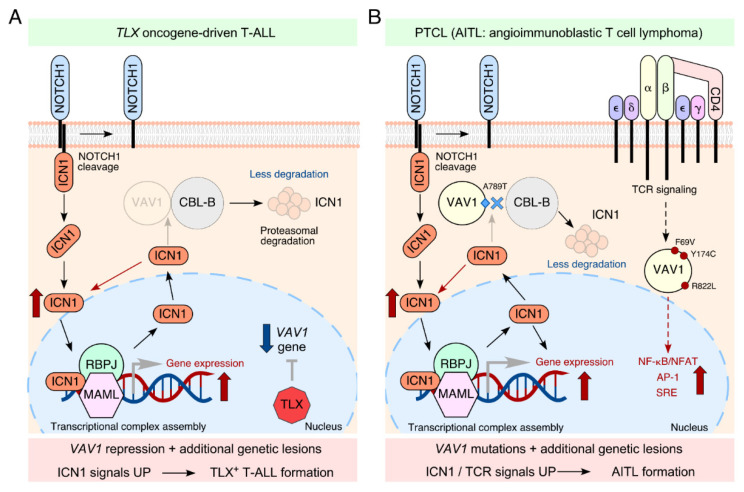
The tumor suppressor activity of VAV1 in T cell neoplasms. (**A**) In human TLX+ T-ALL, the aberrant expression of TLX proteins leads to the transcriptional repression of VAV1 gene and the upregulation of ICN1 signaling. See further details in main text. (**B**) The presence of VAV1 gene mutations that causes the expression of a nonfunctional or absent CSH3 domain might account for the deletion of the tumor suppressor pathway and the increase of ICN1 levels in human PTCLs. We hypothesize that the concurrent engagement of VAV1-dependent canonical routes and ICN1 pathways might play a critical role in T cell transformation. In PTCLs, VAV1 mutations are found at amino acids of the Ac (Tyr^174^), PH (Lys^404^), ZF (Glu^556^) and CSH3 (Arg^798^ and Arg^822^) domains. Furthermore, all the truncations and fusions are targeting the CSH3 domain (i.e. VAV1-MYO1F and VAV1-S100A7). VAV1 mutations in PTLCs are reviewed in [32]. Examples of deficient (unable to bind CBL-B) and gain-of-function (hyperstimulation of RAC1 and NFAT pathways) mutations are shown as blue diamonds and red circles, respectively.

## Data Availability

Not applicable.

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
