# Peer review of "VAV Proteins as Double Agents in Cancer: Oncogenes with Tumor Suppressor Roles"

_biology, 2021, doi:10.3390/biology10090888_

Round 1

Reviewer 1 Report

The review by Cuadrado and Robles-Valero  provides an overview of the dual nature of the Rho family regulator Vav proteins.  These proteins have been originally suggested to have oncogenic roles, however, recently their potential function as tumor suppressors in some tissues have been uncovered.  This is a very timely review, with emerging finding of seemingly contradictory roles for some signalling components. The review is in general well written (suggestions for clarification and some reorganization are below). The text provides a comprehensive and thorough overview of the literature, and discusses exciting new data. In fact, the new data are paradigm shifting not only for the field of vav proteins but  in general for GEF proteins. This fact further augments the importance and timeliness of this review.

The main critique is that the text is not always easy to follow, and some concepts are not emphasized or explained sufficiently. Some sentences are complicated, and the significance of certain details can get lost. See suggestions for improvement below. Importantly, section 4 would benefit from a reorganization to improve the logic and for a clearer definition and classification of some concepts. The detailed critic below provides suggestions for changes that would help the general reader better follow the text.  

Major comments and suggestions

-Summary: The “simple summary” could be simplified. First, it is almost as long as the abstract, and could be made shorter by eliminating some repetitions. The sentence starting at the middle of line 10  (The implication of this family…) is especially complicated. This summary should contain just a few simple statements, to indicate the main message of the review, that in contras to previously defined pro-oncogenic roles, data is emerging suggesting a dual role for Vav proteins.

Introduction: Addition of information on the specificity of the Vav proteins towards Rho family small GTPases would be useful in this section. The authors state that there are both Rho-dependent and independent effects of Vavs, and explain the basis for the Rho-independent effects as consequence of the multitude of proteins binding to various domains.  This however by itself does not necessarily indicate GEF independent functions, since the interactions could serve to localize Rho protein activation (i.e. GEF function of Vavs) to different subdomains or complexes. The concept of Rho (GEF) dependence and independent downstream effects are a key concept, and  should be more explicitly addressed, as in its current form this notion is not clear enough. What is the proof for Rho and GEF-independent effects? Are these clearly established or only hypothesized? It would also be very helpful to indicate for any described effects in further sections whether they are Rho-dependent or not (or indicate if this has not been tested). For example, is the effect of Vav1 on ubiquitination of Notch (section 2) Rho-dependent?

-Please elaborate on the importance of the tyrosine-phosphorylation independence of the effect described in the section starting in 129 (does this indicate independence of Rho GEF functions or at least GEF activity regulation?).

-There are a lot of abbreviations, mostly tumor-specific and signalling component terms used, that the general audience will find difficult to decode. These make it hard to follow some sections of the text. While it is understood that often it is hard to avoid these abbreviations, please provide explanation where this is missing. Sometimes, the explanation comes well after the first use of the abbreviation, or is simply missing at the first use.  For example, please explain what TLX proteins are (line 139); PTCL (line 161) and HNRNPK (line 188).

-Section 4: the authors discuss Vav2 and 3. It would be helpful to remind the reader at the beginning of this chapter of the differences (specificity, tissue expression) between Vav1 and 2/3. Is Vav1 present in tissues where Vav 2 and 3 are expressed?

As mentioned before, this section could benefit from some reorganization. The logic could be improved by explicitly stating some concepts.  First, the section  title implies that it  will discusses the idea that Vav2 and 3 might have roles that are similar to Vav1 (I.e. tumor suppressor roles), but the section starts by exploring the connection between the effects of Vav2 and 3. It might be useful to start by stating in general differences and/or similarities between Vav1 roles (as described in previous sections) and Vav2/3. Next, the section elaborates on data (e.g. for skin tumors) that suggest a GEF-dependent role, which seems different of those described for Vav1. This should also be explicitly stated. The next part discusses data that indicates a pro carcinogenic role for Vav2, which is again in contrast to the emphasis on the tumor suppressive role discussed earlier for vav1. It  would be useful to move the beginning of the last paragraph (i.e. a couple of sentences starting from line 216 should) up, before the discussion of the carcinogenic role of Vav2. This will provide more clarity for the reader.

-Final remarks: In line 261 the authors refer to the canonical oncogenic function of Vav proteins. However, it is not clear what they mean by this? Is this a reference to the GEF function (Rhi dependent downstream effects)? Please clarify. The definition of the canonical roles might be useful to explain early in the introduction, where they discuss Rho dependent and independent effects. Thus,  defining a “canonical” (GEF? ) role and the non-canonical effects (which is the main  topic of this review) early could increase clarity.

Minor comments

-End of line 201: “ As a result only the expression of both proteins could rescue these defects…” It is not clear what defects the authors refer to. Please clarify.

-The word “therefore” in the sentence at the end of line 117 is used incorrectly, as the following statement does not logically follow from that statement. It could be more adequate to use “Since” instead “therefore”, or remove it entirely.   

-Line 151: was the unexpected Vav expression found in pancreas and lung cancer, or normal tissues? Please clarify (add the word cancer if needed), as this is relevant to the oncogenic vs tumor suppressor effects

-Line 240: please add the full name of the mutant after Gly17 (RhoG17V) to make the later reference to this easier to understand.

-Line 245: “suggest the implication of these proteins” should be replaced by  ”implicates these proteins in..”.

Reviewer 2 Report

The review by Cuadrado and Robles-Valero is a well-written succinct manuscript on a very interesting topic on the VAV family of guanine nucleotide exchange factors that are involved in the activation of monomeric G-proteins. It summarizes an interesting new area. While loss-of-function mutations of GAPs that help to inactivate the G-proteins drive some cases cancers, the tumor suppressive role of GEFs is extremely novel. This review should interest a wide range of researchers.

The authors may want to expand or add to the figures to include more of the proteins described in the text.

Reviewer 3 Report

The review by Myriam Cuadrado and Javier Robles-Valero focuses on emerging tumor suppressor functions of VAV proteins with focus on VAV1 in T-ALL. RHO GEFs are typically associated with tumorigenesis. However, VAV1 can act as a tumor suppressor by regulating the levels of the active form of NOTCH1 (ICN1). The authors argue that the SH3 domains of VAV1 bind to CBL-B regardless of VAV1 activation status and this interaction regulates the levels of ICN1 by ubiquitination followed by its degradation. In TLX-driven T-ALL, VAV1 expression is repressed leading to the accumulation of ICN1 and TLX+ T-ALL formation. Similarly, VAV1 mutations that inhibit CBL-B binding increase ICN1 signaling in PTCLs.

Next, the authors argue for a putative tumor suppressor role for the homologous VAV2 and VAV3. The available in vivodata for VAV2 contradict this hypothesis. The authors note that VAV3, which is rarely mutated in human cancer can be instead transcriptionally repressed. Finally, the authors discuss antitumorigenic mutations for RHO GTPases and their GEFs and other outstanding questions in the field.

Overall, the review is well-written and clear. The figures are very comprehensible. The review touches on an area of signaling that is usually neglected as most in the field promote an oncogenic role for RhoGEFs.

The major weakness of the review is that the authors refer on several occasions to their own work, which was published in 2017 (ref #12) and the review reads as if they are reviewing their own work, which defeats the purpose of the review. The authors also refer to a paper they submitted for publication or to unpublished data (lines 196, 309, and 250).

One point needs clarification. As the authors noted and nicely depicted in Figure 1B, CBL-B interacts with the SH3 domains of VAV1 under stimulated and non-stimulated conditions. This would lead to continuous ICN1 degradation and therefore no ICN1 activity if this were true. Can the authors comment on or clarify how the CBL-B/VAV1 interaction is regulated?

A few suggestions:

  • Line 16: we focus on instead of ‘we pay attention to’.
  • Line 61 and 64 and rest of text: change “closed” when describing the inactive conformation of VAV1 to autoinhibited. Closed is not accurate and doesn’t mean much.
  • What is the incidence of VAV1 mutation in human T-ALL and where are they localized in the sequence?
  • Line 210: introduce (HNRNPK) right after ribonucleoprotein K
  • Line 121-122: The last sentence is inaccurate. If there is no data supporting the involvement of other RhoGEF besides Vav1, the authors should replace RHO GEFS with Vav1 or ‘one Rho GEF’.
  • Line 134: Isn’t it ‘types of tumors’?
  • Figure 1A: The authors should specify in the figure legend that the numbering is for VAV1.
  • Figure 1A: In the lilac and yellow boxes ‘GEF-dependent and -independent functions’ and not function.
  • Figure 1B: pY541 and pY544 (red circles) look like they are part of the PH domain in the drawing on the right, but they are not. Please move them.
  • Figure 2: instead of left and right panel use A and B. Right panel, indicate at least one VAV1 mutation. Can the authors specify relevant VAV1 mutation(s) in the context of PTCL.

Round 2

Reviewer 1 Report

Thank you to the authors for addressing my comments. All my concerns have been thoroughly addressed.

Reviewer 3 Report

In their revised manuscript, the authors addressed the points I raised in my review. However, the revised version contains typos that must be corrected.

  • Figure 2: the figure legend still has ‘Left panel’ and ‘Right panel’ while the panels are labeled A and B. Replace left panel with A and right panel with B.
  • Line 210: If the VAV1 missense mutations are found in the Ac, PH, ZF, and CSH3 domains, they are not clustered. Replace ‘VAV1 missense mutations are mainly clustered’ by ‘VAV1 mutations are found at amino acids…’
  • Line 212: what does the sentence ‘VAV1 break the gene’ mean? Aren’t they VAV1 mutants?
  • Line 213: Replace with ‘VAV1 mutations in PTLCs are reviewed in [35]’
  • Line 214 states blue and red diamonds but panel B shows one blue diamond and red circles. Please clarify.
  • Line 419: significant progress has been made and not have been made.
